# Eye movements reflect memory-related theta activity in the human brain

Humza N. Zubair[1,2☉], Matthias Stangl[3,4☉], Uros Topalovic[5], Cory Inman[6], Martin Seeber[5], Sonja Hiller[2], Vikram R. Rao[7], Casey H. Halpern[8], Dawn Eliashiv[9], Itzhak Fried[2,10,11], Nanthia Suthana[5,12]*

1 Medical Scientist Training Program, David Geffen School of Medicine, University of California Los Angeles, Los Angeles, California, United States of America, 2 Department of Psychiatry and Biobehavioral Sciences, Jane and Terry Semel Institute for Neuroscience and Human Behavior, University of California Los Angeles, Los Angeles, California, United States of America, 3 Department of Biomedical Engineering and Department of Psychological & Brain Sciences, Center for Systems Neuroscience, Cognitive Neuroimaging Center, Neurophotonics Center, Boston University, Boston, Massachusetts, United States of America, 4 Department of Neurosurgery, Boston Medical Center, Boston University Chobanian and Avedisian School of Medicine, Boston, Massachusetts, United States of America, 5 Department of Neurosurgery, Duke University, Durham, North Carolina, United States of America, 6 Department of Psychology, University of Utah, Salt Lake City, Utah, United States of America, 7 Department of Neurology and Weill Institute for Neurosciences, University of California San Francisco, San Francisco, California, United States of America, 8 Department of Neurosurgery, University of Pennsylvania, Philadelphia, Pennsylvania, United States of America, 9 Department of Neurology, University of California Los Angeles, Los Angeles, California, United States of America, 10 Department of Neurosurgery, David Geffen School of Medicine, University of California Los Angeles, Los Angeles, California, United States of America, 11 Faculty of Medicine, Tel Aviv University, Tel Aviv, Israel, 12 Department of Biomedical Engineering, Duke University, Durham, North Carolina, United States of America

☉ These authors contributed equally to this work.
* nanthia.suthana@duke.edu

## Abstract

Numerous studies across species emphasize the importance of theta oscillations within medial temporal lobe (MTL) regions, such as the hippocampus, in relation to memory. In rodents, physical movement strongly influences theta activity, while this relationship remains more ambiguous in primates. This disparity could stem from the increased reliance on visual search in primates during navigation. To explore this, we analyzed intracranial electroencephalographic (iEEG) activity from the human MTL recorded simultaneously with body and eye movements during ambulatory navigation. We found that MTL theta power was significantly higher during periods when saccadic eye movements were taking place, and this effect was observed only during periods with overt memory demands. The largest increases occurred during saccades with more variable and exploratory gaze patterns, on trials with better memory performance, and during the early planning period of each route. The modulation was also amplified near environmental boundaries, spatial features known to anchor memory representations and guide navigation. During memory-guided navigation, theta power further tended to increase during both locomotion and stationary periods, consistent with broad engagement during active information gathering. In addition

**Data availability statement:** The data supporting the findings of this study are openly available via Zenodo at https://doi.org/10.5281/zenodo.18487389.

**Funding:** This work was supported by the McKnight Foundation (https://www.mcknight.org), the W. M. Keck Foundation (https://www.wmkeck.org), and the National Institutes of Health (NIH; https://www.nih.gov) under grants U01NS103802 and U01NS117838 (to N.S.), K99NS126715 (to M.S.), and T32GM008042 and F30MH138086 (to H.Z.). H.Z. also received support from the David Geffen Scholarship at UCLA (https://medschool.ucla.edu). The funders had no role in study design, data collection and analysis, decision to publish, or preparation of the manuscript.

**Competing interests:** The authors have declared that no competing interests exist.

**Abbreviations**: CT, computed tomography; ERP, event-related potential; FDR, false discovery rate; HP, high-performance; iEEG, intracranial electroencephalographic; LP, low-performance; MP, medium-performance; MTL, medial temporal lobe; PPC, pairwise phase consistency.

to these memory-specific effects, theta aligned its phase to saccade onset during both memory-guided and visually-guided navigation, suggesting that eye movements impose a consistent temporal structure on ongoing MTL activity. Together, these findings reveal that memory-related theta dynamics in the human MTL are tightly coupled to exploratory visual search and prospective planning during memory-guided navigation, revealing a mechanism by which saccades may help organize mnemonic computations in naturalistic settings.

## Introduction

Theta oscillations (~5–8 Hz) within medial temporal lobe (MTL) regions, including the hippocampus, play a central role in episodic memory and spatial navigation across species [1–6]. While theta activity in rodents is strongly modulated by physical movement, studies in humans have reported a stronger association with memory processes [7]. This apparent discrepancy may arise from methodological differences: rodent studies typically involve recordings during free movement, whereas human studies often rely on stationary, view-based virtual navigation paradigms [8]. As a result, the contributions of memory and movement to theta dynamics in humans remain difficult to disentangle. Ideally, resolving this question requires recording neural activity from deep brain structures like the MTL during naturalistic, ambulatory behavior. Yet, such data remain rare due to the technical challenges of capturing intracranial signals while participants are actively moving.

Recent technological advancements have enabled mobile intracranial electro-encephalographic (iEEG) recordings from the MTL in humans, enabling concurrent investigation of body movements [9]. These techniques have confirmed the presence of movement-related hippocampal theta activity in humans [10]. However, the manifestation of these oscillations varies across human and non-human primate studies [1,11–13]. A potential explanation for this inconsistency could lie in the contrasting modes of exploration employed by primates and rodents. While rodents primarily rely on tactile exploration, particularly through their whiskers, human and non-human primates heavily depend on their well-developed visual system to navigate their surroundings [12,14,15]. Understanding how visual exploration shapes MTL theta is therefore essential for interpreting cross-species differences. Humans further offer the unique advantage of following rapidly shifting task instructions, allowing the effects of movement, visual exploration, and memory demand to be examined within a single experimental framework [8].

In this study, we analyzed mobile iEEG recordings that were simultaneously captured with both body and eye movements as human participants engaged in a real-world ambulatory spatial navigation task. Through alternating task conditions, we systematically modified instructions to modulate ongoing memory demands. Our findings highlight a significant association between MTL theta activity and saccadic eye movements, but only when navigation relied on internal memory retrieval. The largest increases occurred during saccades with more variable and exploratory gaze

patterns and on trials with better memory performance, as well as during the early planning period of each route, when participants generated internal predictions about potential target locations. These modulations were further amplified near environmental boundaries, spatial features that anchor memory representations. During memory-guided navigation, theta power also tended to rise during both locomotion and stationary periods, consistent with broad engagement during active information gathering. In addition, theta exhibited phase resetting to saccade onset during both memory-guided and visually guided navigation, indicating that eye movements may impose a consistent temporal structure on ongoing MTL activity regardless of memory demand. Together, these results indicate that human MTL theta is tightly coupled to exploratory visual search and prospective planning during memory-guided navigation, providing a framework for reconciling variable findings across species and experimental paradigms.

## Results

### Saccades modulate MTL theta activity

We analyzed mobile iEEG recordings in five individuals who were surgically implanted with the RNS System (NeuroPace, Mountain View, CA) in the MTL for the treatment of chronic epilepsy (Fig 1A and 1B) [9,16]. Alongside the iEEG signals, we examined eye and body movements captured while participants were engaged in a spatial navigation task within a rectangular room (Figs 1C and S1A). During the spatial navigation task, participants alternated between two conditions: 1) Visually-cued navigation, where participants walked toward one of 20 distinct wall-mounted signs and 2) Memory-cued navigation, where participants recalled and navigated to previously learned hidden target locations within the room (see Materials and methods).

To determine the impact of eye movements on MTL activity during ambulatory spatial navigation, we computed power across a wide frequency spectrum (2–80 Hz) at all recorded timepoints, comparing periods with saccadic eye movements to fixations (see Materials and methods). Channel-level statistics across the five participants were computed using the Permutation Analysis of Linear Models (PALM) framework [17], which constrains permutations within participants. This approach maintains appropriate statistical independence while preventing effects from being driven by any single individual. Because PALM operates at the level of individual channels while preserving participant structure, we report channel-level analyses throughout the manuscript. Using this approach, we observed a significant increase in MTL power within the low-frequency range (~3–15 Hz) during saccades compared to fixation periods when aggregating across channels ($N_{channels} = 16$; $t = 3.71$, $p < 0.001$; Fig 2A and 2B). Additionally, we found that low-frequency power rose gradually following saccade onset, with statistically significant increases beginning 16 ms post-onset ($N_{channels} = 16$; $p < 0.05$; Fig 2C and 2D). Saccade duration varied substantially (median: 64 ms; interquartile range: 52 ms; $N_{saccades} = 29,747$), with most lasting less than 100 milliseconds (S1B Fig).

### Saccade-related theta activity depends on memory

Given the established link between memory and MTL theta oscillations during spatial navigation [2,7,18,19], we examined how task demands influenced saccade-related oscillatory activity. Memory-cued navigation required participants to encode and retrieve target locations, processes intrinsically linked to theta function [18,20], whereas visually-cued navigation relied on continuous visual access to the target, minimizing memory demands. As previously reported [16], MTL low-frequency power was elevated during memory-cued compared to visually-cued navigation, particularly in the 2–3.75 Hz and 6.75–17.5 Hz ranges ($N_{channels} = 16$; $p < 0.05$; S4A Fig). We next investigated whether this task-dependent difference in MTL oscillatory activity could be driven by eye movements, given their central role in visual processing. As summarized in S1 Table, participants completed the same number of trials across conditions, and the number of saccades and fixations were also identical. Interestingly, saccade-related increases in MTL theta power were significantly more pronounced during memory-cued than visually-cued navigation, specifically within the 5.75–7.5 Hz range (S4B Fig). As a result, our subsequent analyses focused on the 5–8 Hz frequency range. Indeed, we observed that this MTL theta band

PLOS Biology

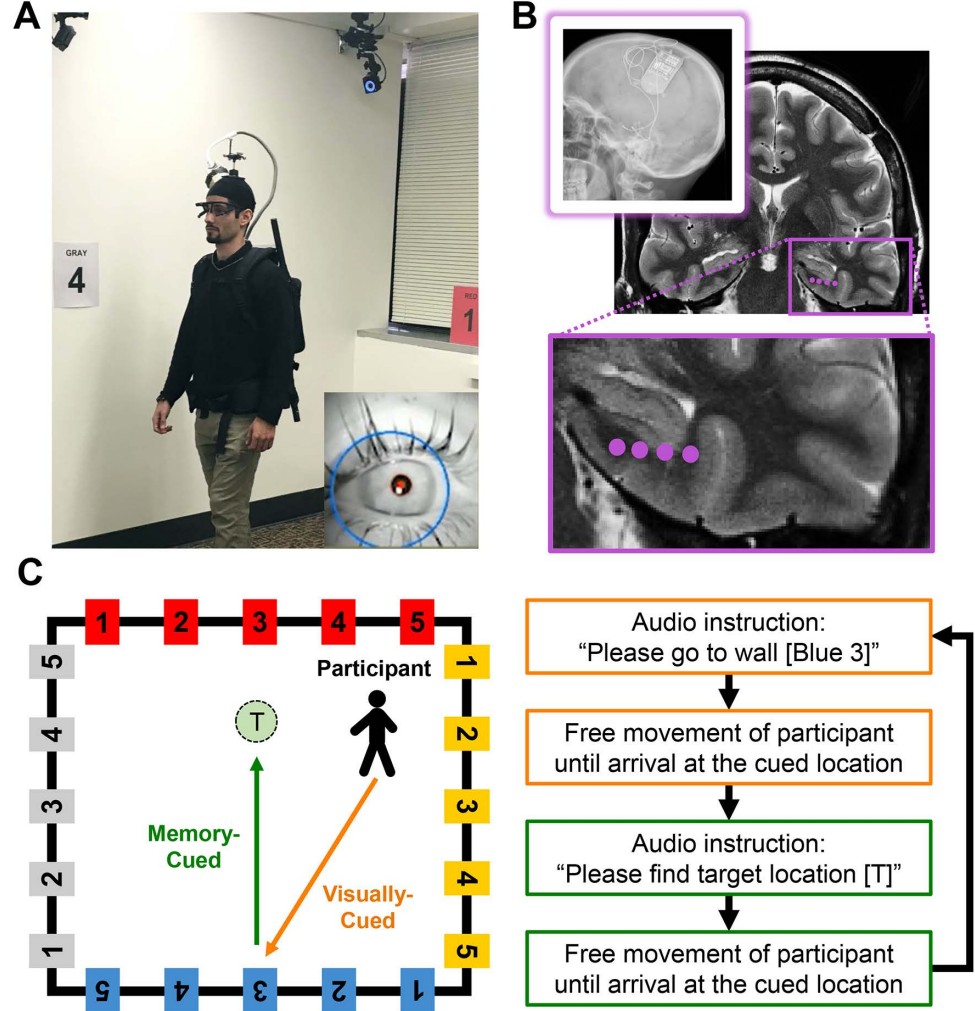

**Fig 1. Experimental design and task. (A)** Intracranial electroencephalographic activity, eye and body movements were recorded as participants freely walked around the room. Wall-mounted motion-tracking cameras recorded the position of on-body reflective markers. Participants also wore an eye-tracking headset to monitor saccadic eye movements. A snapshot from the eye-facing camera is shown in the bottom right. For illustrative purposes, an experimenter is shown wearing the full setup. **(B)** MRI of an example participant with an implanted RNS System. Purple dots indicate the location of four electrode contacts in the left medial temporal lobe (MTL). The top left inset shows an X-ray used to localize electrode positions. **(C)** The environment contained 20 visible wall-mounted signs and three invisible circular target locations (0.7 m diameter). At the start of the task, participants freely explored the room to locate the invisible targets; each time a target was reached, an auditory tone signaled success, allowing them to gradually learn and remember these locations through experience. The task then alternated between two conditions: "visually-cued" navigation, during which participants navigated to a wall-mounted sign (e.g., "Blue 3"), and "memory-cued" navigation, during which they recalled and navigated to the previously learned invisible targets (e.g., "T").

power exhibited an increase during memory-cued navigation, primarily during saccades, in contrast to fixation periods ($N_{channels} = 16$; $t = 2.37$, $p = 0.012$; Fig 3A). This increase unfolded gradually *after* saccade onset, reaching statistical significance after 48 ms ($p < 0.05$; Fig 3B and 3C). In contrast, no significant theta modulation by saccades was observed during visually-cued navigation ($N_{channels} = 16$; $t = 0.31$, $p = 0.376$; Fig 3D–3F). Direct comparison be*tween* conditions confirmed that saccade-related power was significantly greater during memory-cued versus visually-cued navigation ($N_{channels} = 16$; $t = 2.16$, $p = 0.019$; Fig 3G). S1 Video illustra*tes* trial-level data from a representative participant, showing enhanced theta activity during saccades in memory-cued but not visually-cued navigation.

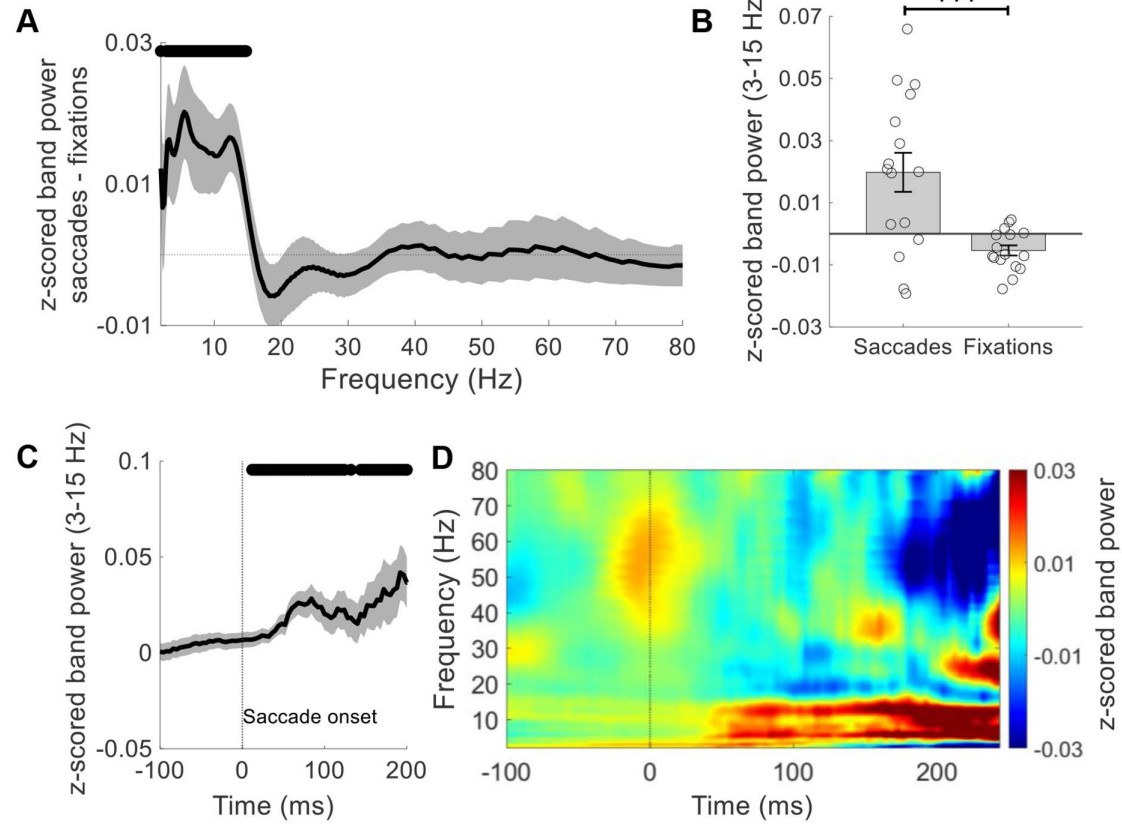

**Fig 2. Saccadic eye movements are associated with increases in MTL low-frequency power. (A)** Normalized (z-scored) band power differences between saccades and fixations across frequencies (2–80 Hz). Positive values indicate higher power during saccades. A significant increase was observed in the 2.75-14.75 Hz range (black horizontal bar = $p < 0.05$). **(B)** Mean low-frequency (3–15 Hz) power was significantly greater during saccades compared to fixations (*** = $p < 0.001$). Circles represent individual MTL channels ($N = 16$); error bars show ±SEM. **(C)** The increase in low-frequency power emerged gradually following saccade onset (dotted vertical line at time = 0) with statistically significant elevations following saccade onset. Black horizontal bars = $p < 0.05$. **(D)** Heatmap showing normalized power over time and frequency, aligned to saccade onset (time = 0). A clear post-saccade increase was observed in the 3-15 Hz band. Warmer colors reflect higher power values. Normalization (z-scoring) was based on mean theta power across all fixation and saccade periods pooled across conditions, ensuring that contrasts were not biased by short pre-saccadic windows. Shaded gray regions in (A) and (C) represent ±SEM across channels ($N_{channels} = 16$). The data underlying this Figure are available here: https://doi.org/10.5281/zenodo.18487389.

To determine whether differences in saccadic modulation of theta power could be explained by variations in eye movement behavior across conditions, we compared key oculomotor metrics between memory-cued and visually-cued navigation (S1D Fig). Specifically, we examined saccade frequency, duration, displacement, and fixation duration. No significant differences were found between the two task conditions in any of these measures ($N_{participants} = 5$; saccade frequency: memory-cued = 2.4 ± 0.2 Hz, visually-cued = 2.3 ± 0.2 Hz, $t = 1.12$, $p = 0.15$; saccade duration: memory-cued = 89.0 ± 3.7 ms, visually-cued = 88.8 ± 3.6 ms, $t = 0.47$, $p = 0.47$; saccade displacement: $t = −0.53$, $p = 0.31$; fixation duration: 342.1 ± 30.8 ms versus 370.8 ± 32.1 ms, $t = −1.20$, $p = 0.18$). These results indicate that the observed saccade-related theta modulation cannot be attributed to differences in eye movement behavior across task conditions.

## Saccades, not body movement, predict memory-related theta activity

Prior studies have shown that MTL theta power can be influenced by variables such as proximity to environmental boundaries [1,16] and walking speed [10]. To disentangle the effects of eye and body movement, we applied linear mixed-effects

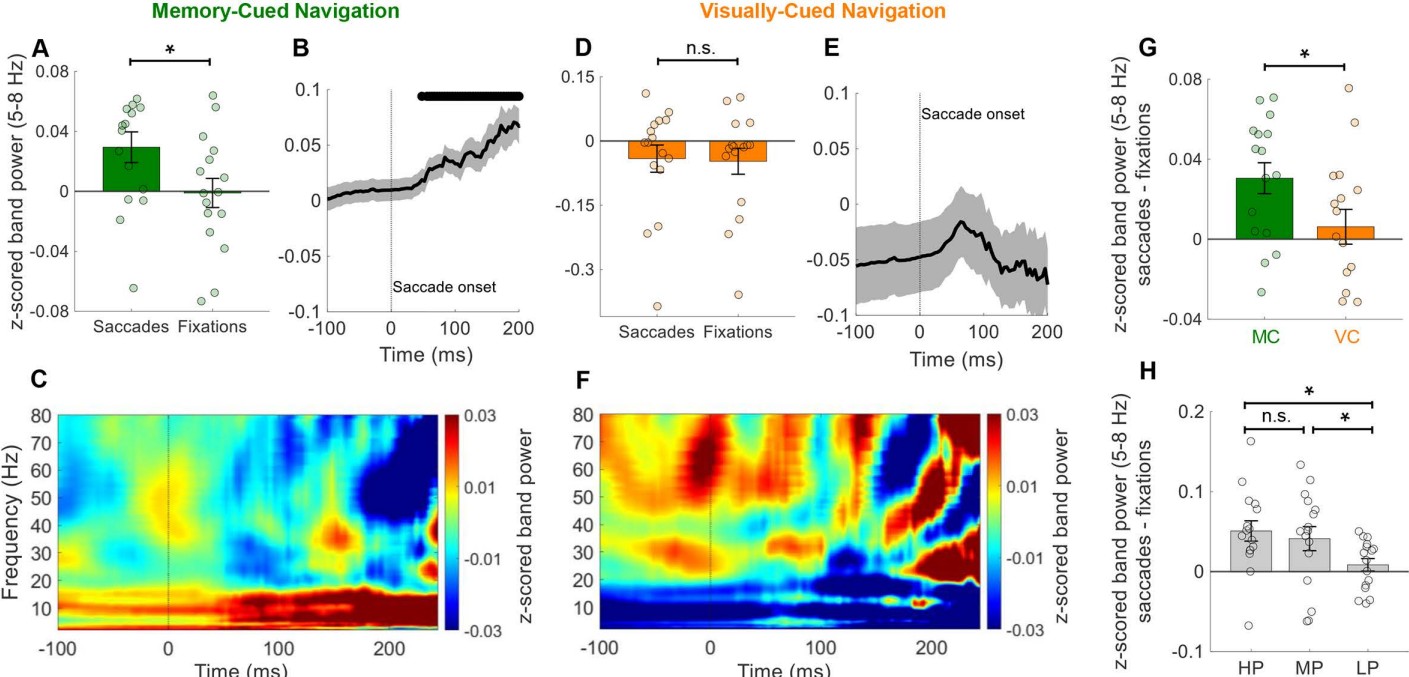

**Fig 3. Saccade-related theta activity depends on memory demands. (A)** During memory-cued (MC) navigation, MTL theta (5-8 Hz) power was significantly higher during saccades compared to fixations ($p = 0.012$). **(B)** Theta power gradually increased after saccade onset (time = 0), with significant elevations beginning at 48 ms (black bars = $p < 0.05$). **(C)** Heatmap shows sustained low-frequency power increases following saccade onset (time = 0). **(D)** During visually-cued (VC) navigation, MTL theta (5–8 Hz) power did not significantly differ between saccades and fixations ($p = 0.376$). **(E)** No significant theta increases were observed after saccade onset (time = 0). **(F)** Heatmap shows no saccade-related increases in low-frequency power, but peaks in gamma (45–80 Hz) power appeared around saccade onset (time = 0). **(G)** Saccade-related theta power (saccades – fixation difference) was significantly greater during MC vs. VC navigation ($p = 0.019$). **(H)** Theta power increases were stronger during high-performance (HP) and medium-performance (MP) MC trials compared to low-performance (LP) trials. Panels A, D, G, and H show mean (± SEM) across 16 MTL channels (circles). For B and E, shaded gray = standard error of the mean (SEM) across channels ($N_{channels} = 16$). Warmer colors in C and F reflect higher power values. * = $p < 0.05$, n.s. = not significant. The data underlying this Figure are available here: https://doi.org/10.5281/zenodo.18487389.

models separately for memory- and visually-cued navigation. In the memory-cued condition, eye movement speed significantly predicted theta power ($N_{participants} = 5$; $t = 3.15$, $p = 0.032$), whereas body movement speed did not ($N_{participants} = 5$; $t = -0.97$, $p = 0.818$; S4C Fig). In contrast, during visually-cued navigation, neither eye nor body movement speed significantly modulated theta power ($N_{participants} = 5$; $t = 0.10$, $p = 0.436$ and $t = -0.31$, $p = 0.656$, respectively).

We next examined whether saccade-related theta increases were modulated by other behavioral contexts. We quantified z-scored theta power during saccades across four contexts: near boundaries, in inner regions, at higher locomotion speeds (movement), and while stationary (S5A–S5H Fig). Saccade-related theta increases were significant only near boundaries during memory-cued navigation ($N_{channels} = 16$, MC: $t = 2.62$, $p = 0.006$; VC: $t = -0.02$, $p = 0.506$; MC versus VC: $t = 2.51$, $p = 0.008$; S5A and S5B Fig). In contrast, no significant effects were observed in inner regions for either condition ($N_{channels} = 16$, MC: $t = 1.00$, $p = 0.163$; VC: $t = 0.66$, $p = 0.163$; MC versus VC: $t = 0.45$, $p = 0.324$; S5C and S5D Fig). During movement and stationary periods, effects trended towards significance only during memory-cued navigation (movement: MC: $t = 1.60$, $p = 0.060$; VC: $t = 0.10$, $p = 0.462$; MC versus VC: $t = 1.70$, $p = 0.051$; S5E and S5F Fig; stationary: MC: $t = 1.35$, $p = 0.095$; VC: $t = 0.14$, $p = 0.440$; MC versus VC: $t = 1.62$, $p = 0.058$; S5G and S5H Fig). Taken together, these results show that the observed increases are not explained by locomotion or general movement state and instead appear most robust near boundaries, where spatial information is especially salient for memory-guided navigation.

**Saccade-related theta activity is strongest during successful and early phases of memory-guided navigation**

Given that saccadic modulation of MTL theta power was observed only during memory-cued navigation, we investigated whether this effect varied with memory performance. Behavioral performance was quantified by the walking detour distance (error) on each memory-cued navigation trial (see Materials and Methods). Saccade-related MTL theta power was significantly higher in high-performance (HP) and medium-performance (MP) trials compared to low-performance (LP) trials ($N_{channels} = 16$, HP versus LP: $t = 2.40$, $p = 0.027$, MP versus LP: $t = 2.09$, $p = 0.032$, HP versus MP: $t = 0.34$, $p = 0.373$, false discovery rate (FDR) corrected, Fig 3H). Furthermore, the difference in theta power between saccades and fixations reached significance only in HP and MP trials ($N_{channels} = 16$, HP: $t = 3.68$, $p = 0.001$; MP: $t = 3.26$, $p = 0.003$), but not in LP trials ($N_{channels} = 16$, $t = 1.34$, $p = 0.095$). Importantly, these theta power differences were not driven by changes in eye movement behavior, as saccade duration, frequency, displacement, and fixation duration showed no significant differences across performance groups (all $p > 0.1$; S1E Fig). Together, these findings indicate that memory-related MTL theta activity in humans is closely linked to saccadic eye movements, with this relationship being strongest when memory performance is successful.

To further examine the temporal dynamics of theta activity within memory-guided navigation, we divided each memory-cued trial into an early ("planning") and late ("execution") half. The first half corresponded to initial route planning and search behavior, while the second half reflected execution of the recalled path toward the hidden target. Theta power during saccades was significantly higher than during fixations in the first half ($N_{channels} = 16$, $t = 2.29$, $p = 0.013$; S2A Fig), but not in the second half ($N_{channels} = 16$, $t = 0.68$, $p = 0.254$; S2B Fig). The difference in saccade–fixation theta modulation between halves was significant ($N_{channels} = 16$, $t = 1.79$, $p = 0.041$; S2C Fig). In contrast, visually-cued navigation showed no significant saccade-fixation modulation in either half: first half ($N_{channels} = 16$, $t = -0.120$, $p = 0.546$; S2D Fig), second half ($N_{channels} = 16$, $t = 0.459$, $p = 0.327$; S2E Fig), or their difference ($N_{channels} = 16$, $t = -0.761$, $p = 0.772$; S2F Fig). We also confirmed that movement speed increased from planning to execution, consistent with behavioral expectations. Velocity was significantly higher in the second half than the first half for both memory-cued navigation ($N_{participants} = 5$, $t = -8.400$, $p = 0.030$; S2G Fig) and visually-cued navigation ($N_{participants} = 5$, $t = -13.513$, $p = 0.031$; S2H Fig), indicating that this temporal modulation of theta is not explained by changes in movement alone. Together, these findings show that saccade-related MTL theta activity is strongest during the early, planning-dominant portion of memory-guided trials and during successful retrieval, supporting a role for theta in anticipatory visual search and memory-driven planning rather than low-level motor execution.

To assess whether observed saccade-related theta effects could be explained by systematic covariation among behavioral predictors, we directly examined relationships between trial phase, spatial context, performance, and eye movement metrics during memory-cued navigation. As expected from task structure, memory-cued trials typically began near the boundary, immediately following visually-cued navigation. Accordingly, participants spent a greater proportion of time near the boundary during the first compared to the second half of memory-cued trials ($N_{channels} = 16$; $t = 3.91$, $p = 0.032$). Saccade duration was also modestly longer during the first versus second half ($N_{channels} = 16$; $t = 2.92$, $p = 0.032$). These results indicate that early trial phase and boundary proximity are partially coupled in this task. Importantly, no other behavioral measures differed across these contrasts. Eye speed, scanpath entropy, and boundary occupancy did not differ across performance levels (all $p > 0.19$). Eye speed and scanpath entropy also did not differ between early and late trial phases (all $p > 0.25$), and saccade frequency, saccade displacement, and fixation duration were comparable between the first and second halves of memory-cued navigation (all $p > 0.15$). When comparing boundary and inner regions of the room, eye speed and scanpath entropy did not differ (all $p > 0.09$). Together, these analyses indicate that while early trial phase and boundary proximity co-occur due to task structure, the broader pattern of saccade-related theta modulation cannot be explained by a single behavioral confound or shared covariance among eye movement parameters, consistent with a role for internally driven planning processes that are most prominent early in memory-guided navigation.

PLOS Biology

## MTL theta activity reflects exploratory eye movements during memory-guided navigation

We next examined how MTL theta activity relates to the magnitude, frequency, and spatial complexity of saccadic eye movements. Larger or more exploratory saccades may reflect greater shifts in visual sampling or planning effort during memory retrieval.

**Saccade magnitude.** Saccades were categorized by magnitude using a median split, based on the two-dimensional displacement from saccade onset and termination. MTL theta power was significantly higher during large versus small saccades ($N_{channels}$ = 16, $t$ = 3.28, $p$ = 0.002; Fig 4A). When analyzed by condition, this effect was present only during memory-cued navigation ($N_{channels}$ = 16, $t$ = 2.72, $p$ = 0.004) and absent during visually-cued navigation ($N_{channels}$ = 16, $t$ = 0.85, $p$ = 0.204; Fig 4B). The magnitude-related increase in theta power was also significantly greater in memory-cued than visually-cued navigation ($N_{channels}$ = 16, $t$ = 2.20, $p$ = 0.020; Fig 4C). Across performance levels, theta power during large saccades was numerically higher in high-performance (HP) than in medium (MP) or low-performance (LP) trials, though differences did not survive FDR correction ($N_{channels}$ = 16, *HP versus MP*: $t$ = 2.29, $p$ = 0.047; *HP versus LP*: $t$ = 1.69, $p$ = 0.071; *MP versus LP*: $t$ = −1.02, $p$ = 0.845; FDR corrected; Fig 4D). These results indicate that saccade displacement magnitude modulates theta power specifically during memory retrieval, consistent with enhanced neural engagement during larger exploratory gaze shifts.

**Saccade frequency.** We next assessed whether theta power varied by saccade frequency. One-second epochs were median-split by saccade count within each participant and condition. During memory-cued navigation, MTL theta power was trending higher during high- versus low-saccade periods ($N_{channels}$ = 16, $t$ = 1.33, $p$ = 0.096; S4D Fig – Left), while no significant difference emerged during visually-cued navigation ($N_{channels}$ = 16, $t$ = −0.34, $p$ = 0.630; S4D Fig – Right). The difference between high and low frequency epochs was greater for memory-cued than visually-cued navigation ($N_{channels}$ = 16, $t$ = 1.72, $p$ = 0.049; S4E Fig). These results suggest that theta power increases with the rate of saccadic sampling, particularly when visual exploration must rely on memory.

**Scanpath entropy.** To quantify the spatial complexity of eye movements, we computed scanpath entropy using a grid-binned spatial occupancy approach [21,22]. The visual field was divided into 30 × 30 bins, and Shannon entropy was calculated within 1-s sliding windows. For each participant, we then performed a median split of entropy values into high- and low-entropy time points and compared the corresponding 5–8 Hz z-scored theta power. In memory-cued navigation, theta power was significantly higher during high-entropy than low-entropy periods ($N_{channels}$ = 16, $t$ = 2.06, $p$ = 0.023; Fig 4E), whereas in visually-cued navigation the difference was not significant ($N_{channels}$ = 16, $t$ = 1.03, $p$ = 0.158; Fig 4F). These results suggests that greater gaze dispersion or more exploratory scanpaths are associated with stronger theta activity, particularly during memory-guided navigation. Together, these results indicate that MTL theta power may track multiple aspects of visual exploration, including saccade magnitude, frequency, and spatial dispersion, during memory-guided navigation. Theta thus appears to reflect internal cognitive processes linked to active search and spatial planning rather than simple oculomotor behavior.

## Saccades induce theta phase resetting independent of memory and trial phase

To test whether saccades elicit phase resetting of MTL theta, we computed pairwise phase consistency (PPC) across saccades [23]. Consistent with prior studies in humans and non-human primates demonstrating saccade-locked theta phase resetting [2,6,24,25], PPC was significantly elevated around saccade onset from –60 to +196 ms relative to saccade onset ($N_{channels}$ = 16, black bar = $p$ < 0.05; S3A Fig). Using a predefined saccade-onset window (−50 to +200 ms) and a baseline period of equal duration (−800 to −550 ms) [6], we confirmed that PPC during saccades was significantly higher than baseline across all navigation conditions ($N_{channels}$ = 16, $t$ = 2.62, $p$ = 0.002; S3B Fig), indicating reliable theta phase resetting associated with saccadic events.

When analyzed separately by task condition, phase resetting occurred in both memory-cued and visually-cued navigation. In memory-cued trials, PPC increased from −72 to +208 ms ($N_{channels}$ = 16, black bar = $p$ < 0.05; S3C Fig) and was

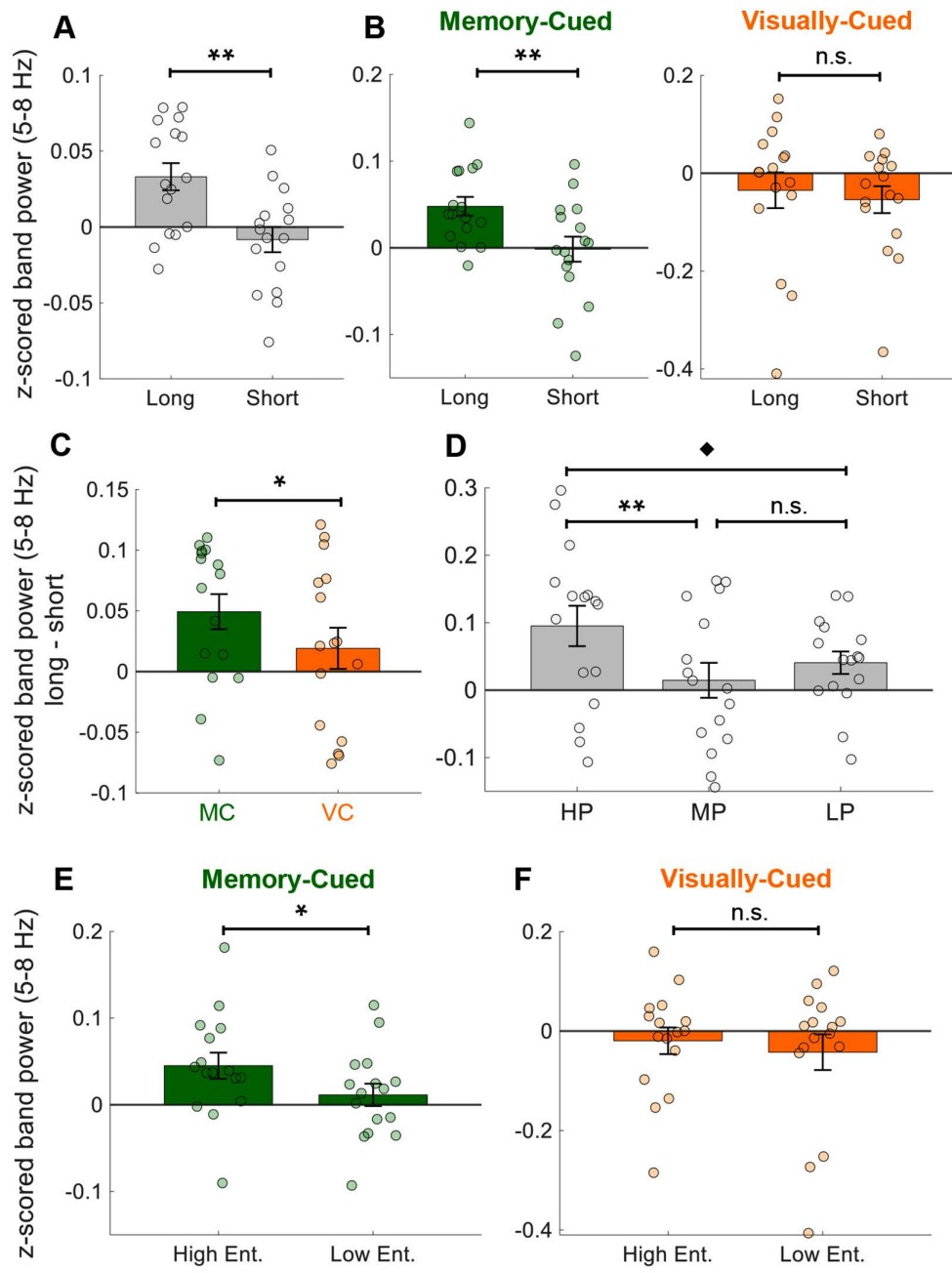

**Fig 4. MTL theta activity scales with saccade magnitude during memory-guided navigation. (A)** MTL theta (5–8 Hz) power was significantly higher during large compared to small saccades ($p = 0.002$). **(B)** This effect was present during memory-cued (MC) navigation ($p = 0.004$) but not during visually-cued (VC) navigation ($p = 0.204$). **(C)** The magnitude-related increase in theta power (difference between large and small saccades) was significantly greater during MC than VC navigation ($p = 0.020$). **(D)** During MC trials, saccade magnitude-related theta increases were stronger in high-performance (HP) trials than in medium (MP) and low-performance (LP) trials. **(E-F)** Mean normalized (z-scored) 5–8 Hz theta power during high- and low-entropy periods for each condition. Theta power was significantly higher during high-entropy periods in MC navigation ($p = 0.023$; **E**) and was not significant in VC navigation ($p = 0.158$; **F**). Panels A-F show mean (± SEM) across 16 MTL channels (circles). ♦ = p < 0.1, * = p < 0.05, ** = p < 0.01, n.s. = not significant. The data underlying this Figure are available here: https://doi.org/10.5281/zenodo.18487389.

significantly higher during saccades than baseline ($N_{channels}$ = 16, $t$ = 2.12, $p$ = 0.006; S3D Fig). In visually-cued trials, a shorter window of significant PPC was observed from +40 to +52 ms (N$_{channels}$ = 16, black bar = $p$ < 0.05; S3E Fig) and PPC during saccades was also significantly higher than baseline (N$_{channels}$ = 16, $t$ = 2.70, $p$ = 0.005; S3F Fig). The presence of phase rese*tt*ing in both memory-cued and visually-cued navigation suggests that saccade-linked theta synchronization is not exclusively memory-dependent but may instead reflect a general mechanism coupling visual sampling with MTL timing. The longer post-saccade duration in the memory-cued condition may point to sustained mnemonic or planning-related processes extending beyond immediate sensory updating.

To assess whether phase resetting varied across trial phases, we divided each trial into early ("planning") and late ("execution") halves. Theta phase resetting was significant in both early ($N_{channels}$ = 16, $t$ = 2.51, $p$ = 0.005) and late ($N_{channels}$ = 16, $t$ = 2.31, $p$ = 0.004) halves (S3G–S3J Fig), suggesting that this synchronization accompanies saccadic events throughout navigation. Together, these findings suggest that saccades serve as temporal anchors for theta phase resetting in the human MTL, supporting both visual exploration and higher-order cognitive operations such as planning and memory retrieval.

### Robustness of results across time–frequency parameters

Time-frequency analyses of electrophysiological data involves a trade-off between temporal and spectral resolution. Because saccades are extremely brief (typically < 100 ms), while theta oscillations are slower (5–8 Hz, corresponding to cycles lasting ~125–200 ms), we aimed to maximize resolution in the time domain for our theta analysis. In our main analyses, we used a six cycle wavelet approach to balance this trade-off [26]. To ensure our findings were not dependent on this choice, we repeated the key analyses using a lower number of cycles (3), which improves temporal resolution at the cost of spectral precision. These control analyses confirmed our main results. Specifically, MTL low-frequency power (3–15 Hz) was significantly elevated during saccades compared to fixations ($N_{channels}$ = 16, $t$ = 3.30, $p$ = 0.001; S6A Fig). This increase remained specific *to* memory-cued navigation ($N_{channels}$ = 16, $t$ = 2.28, $p$ = 0.013) and was no*t* observed during visually-cued navigation ($N_{channels}$ = 16, $t$ = 0.65, $p$ = 0.258; S6B Fig). Wi*th*in memory-cued navigation, saccade-related MTL theta band power was again higher in HP and MP trials compared to LP trials ($N_{channels}$ = 16, *HP versus LP*: $t$ = 2.35, $p$ = 0.012; *MP versus LP*: $t$ = 1.87, $p$ = 0.035; *HP versus MP*: $t$ = 0.35, $p$ = 0.365; FDR correc*t*ed; S6C Fig). Additionally, the saccade-fixation theta difference was significant only for HP and MP trials ($N_{channels}$ = 16, *HP*: $t$ = 3.47, $p$ < 0.001; *MP*: $t$ = 2.75, $p$ = 0.004; *LP*: $t$ = 1.23, $p$ = 0.112).

When comparing high- versus low-entropy periods in MC navigation, theta power was elevated during memory-cued navigation ($N_{channels}$ = 16, $t$ = 2.06, $p$ = 0.023) but not visually-cued navigation ($N_{channels}$ = 16, $t$ = 1.03, $p$ = 0.158; S6D Fig). Similarly, theta power was elevated during long versus short saccades exclusively in memory-cued navigation ($N_{channels}$ = 16, $t$ = 2.71, $p$ = 0.004) but not visually-cued navigation ($N_{channels}$ = 16, $t$ = 0.84, $p$ = 0.205; S6E Fig). Comparing high- with low-frequency saccade periods revealed a trending increase in memory-cued navigation ($N_{channels}$ = 16, $t$ = 1.33, $p$ = 0.096) and no increase during visually-cued navigation ($N_{channels}$ = 16, $t$ = −0.34, $p$ = 0.630; S6F Fig).

Moreover, we confirmed that in boundary regions, MTL theta power was elevated during saccades compared to fixations during memory-cued navigation ($N_{channels}$ = 16, $t$ = 2.60, $p$ = 0.007) but not during VC navigation ($N_{channels}$ = 16, $t$ = −0.02, $p$ = 0.506; S6G Fig). For inner regions of the room, theta power did not differ between saccades and fixations for either memory-cued ($N_{channels}$ = 16, $t$ = 1.16, $p$ = 0.130) or visually-cued navigation ($N_{channels}$ = 16, $t$ = 1.01, $p$ = 0.164; S6H Fig). Finally, comparing theta power between saccades and fixations revealed elevations during the first half of MC trials ($N_{channels}$ = 16, $t$ = 2.29, $p$ = 0.013) but not during the second half ($N_{channels}$ = 16, $t$ = 0.69, $p$ = 0.251; S6I Fig). Overall, these results demonstrate that the observed saccade-related modulation of MTL theta during memory-guided navigation is robust across time-frequency parameters choices.

### Discussion

Our findings demonstrate that during real-world ambulatory navigation, human MTL theta activity is shaped by memory demands and modulated by saccadic eye movements. We observed increased theta power during saccades, but

only during successful memory-cued navigation, when target locations had to be internally recalled. This modulation was absent during visually guided navigation, where the target remained visible and no memory retrieval was required. Importantly, this saccade–theta relationship persisted regardless of the participant's locomotion state, suggesting that eye movements, rather than physical movement, were the primary behavioral correlate of memory-related theta in this context.

A rich literature shows that the relationship between movement and theta differs markedly across species. In rodents, hippocampal theta oscillations in the 5–8 Hz range occur continuously during exploration and are strongly tied to movement speed [27,28]. In contrast, humans, non-human primates, and bats exhibit theta in transient bouts, and its occurrence is less tightly coupled to movement [10,13,16,29–33]. These species differences challenge canonical models that view theta as a movement-driven rhythm supporting memory and navigation [34]. A likely explanation lies in differing sensory strategies. Rodents rely heavily on olfaction and whisking [14,35], whereas humans depend more on vision. In humans, theta may become more aligned with visual sampling through saccades, particularly when memory is engaged. Consistent with this view, rodent studies show theta synchronizing with whisking during memory-guided object discrimination [14]. Our results parallel this pattern, showing that in humans, theta becomes dissociated from movement and instead tracks saccadic eye movements during memory-guided navigation.

Prior work has linked MTL theta to successful memory. Theta power increases during spatial memory retrieval, precedes successful verbal encoding, and is elevated during visual exploration in monkeys when memory is later successful [1–4]. Here, we extend these findings by showing that saccade-related MTL theta power is enhanced during memory-cued compared to visually-cued navigation and is most prominent during higher performance trials. Saccade-related theta amplitude scaled with saccade magnitude, but only in memory-relevant contexts, suggesting that saccades may help structure internal representations by enhancing interregional communication and integrating visual input with stored memories [5,6,12,18,20,36]. This mechanism may support prediction and goal-directed navigation.

We also found evidence of theta phase resetting around saccade onset, consistent with prior work in both humans and non-human primates [2,6,24,25]. Phase resetting was observed in both memory-cued and visually-cued navigation, suggesting that saccade-locked theta serves as a general temporal coordination mechanism in the MTL. However, the phase resetting persisted longer after saccade onset during memory-cued navigation, potentially reflecting additional mnemonic or planning-related processes. This observation supports models proposing that theta phase alignment provides a temporal framework for organizing perception and memory [20,37].

It is important to note that peri-saccadic low-frequency power increases may partially reflect overlapping event-related potential (ERP) components, as reported in prior work [24,25]. Such evoked responses can produce broadband low-frequency power that temporally coincides with theta activity. Nonetheless, our analyses emphasized time–frequency power averaged across many saccades and fixations, rather than single-trial ERP waveforms, and the presence of clear phase resetting (PPC) effects supports the interpretation that rhythmic synchronization contributes meaningfully to the observed signal.

Theta activity was not uniform across the course of a trial. Saccade-related theta power increases were stronger during the early, planning-dominant phase of memory-cued navigation than during later execution. Moreover, periods of higher scanpath entropy, reflecting more exploratory and dispersed gaze behavior, were associated with greater theta power. The link between exploratory viewing and theta echoes prior work showing that hippocampal theta in both humans and monkeys increases during active visual search [2,38]. This pattern suggests that theta may help organize internal predictions or spatial hypotheses before movement, supporting a transition from memory recall to forward-looking planning. In this view, hippocampal theta could act as a bridge between retrieving past experiences and simulating future trajectories, linking perception, attention, and memory into a cohesive planning process.

Previously, we showed that MTL theta is modulated by spatial location, movement speed, and behavioral demands [1,10,16]. In those studies, walking speed was associated with theta increases only when memory was not explicitly engaged. Aghajan and colleagues [10] reported that theta prevalence increased with movement speed during a simple

walking task without memory demands, and Stangl and colleagues [16] found significant modulation of low-frequency power by speed during visually-guided ("no target search") but not memory-cued ("target search") navigation. Using the same dataset as Stangl and colleagues [16], the present study focuses on eye movement–related theta dynamics and similarly finds that movement speed did not significantly explain variance in theta power, whereas eye movements did (S4C Fig). This pattern is consistent with our more recent memory-guided studies [1,39], which also found no relationship between theta and speed under mnemonic conditions. Together, these findings indicate that theta–speed coupling is characteristic of non-mnemonic or visually-guided locomotion, whereas during memory-guided navigation, theta dynamics reflect internal cognitive processes such as retrieval and planning rather than external movement parameters.

In related work, we found that theta increases prior to memory-guided turns, critical moments of route recall, were also associated with behaviorally relevant saccades [39]. Although eye movements explained significant variability in theta during walking, spatial position near moments of recall was a stronger modulator, likely because visual search demands were minimal. In the current task, however, participants used eye movements to search for hidden targets, emphasizing visual exploration during goal-directed search rather than route recall. Given prior evidence of stronger MTL theta near environmental boundaries [16], we also tested whether boundary proximity contributed to the observed effects. Here, saccade-related theta enhancements were significantly greater near boundaries but not within inner regions of the environment. This pattern suggests that boundaries amplify theta engagement during memory-guided exploration, potentially by anchoring spatial predictions or enhancing the retrieval of boundary-referenced representations. Together, these findings suggest that MTL theta flexibly couples to eye movements in memory-guided contexts, particularly when visual search supports the integration of sensory input with internal spatial representations.

Saccades are rapid, coordinated eye movements that play a central role in guiding attention and organizing perception [40]. They are influenced by both bottom-up visual salience and top-down task goals [41–45]. Each saccade interrupts visual input and prompts the realignment of spatial reference frames, integrating egocentric and allocentric representations [13,41,46,47], a process that is cognitively demanding. This momentary disruption may create brief windows of opportunity for memory-related neural computations [2,41,48]. Despite their brevity, typically lasting less than 100 milliseconds, saccades appear to engage neural systems operating at slower timescales, such as theta oscillations, which cycle approximately every 125–200 ms. This temporal mismatch suggests that saccade-related increases in theta may not be tied to individual saccades per se, but instead to broader cognitive states or sequences of saccadic activity.

Several possible mechanisms could explain the observed relationship between saccadic eye movements and theta activity. Theta may be elevated during bursts or clusters of saccades, as individuals scan the environment during memory retrieval. Alternatively, theta may reflect a cognitive state associated with saccade planning, such as attentional engagement or retrieval initiation, rather than the saccadic movements themselves. It is also possible that theta is suppressed during sustained fixations, with saccades marking moments of re-engagement or attentional reset. Another possibility is that saccades act as cognitive boundaries, triggering transient theta bursts that help segment experience and organize information in memory. Together, these possibilities support a view in which saccades serve not only as perceptual and motor events but also as functionally meaningful anchors for memory-related activity in the MTL.

The hippocampus and broader MTL are well-positioned to support this integration. Corollary discharge signals from oculomotor control regions provide information about saccade direction and amplitude to MTL circuits [5,12,24]. This supports anticipatory visual coding and aligns with our finding that larger saccades induce greater internal map transformations [36,49]. Consistent with this, we found that saccade magnitude is associated with increased theta amplitude during memory retrieval. In contrast, this relationship was absent during visually guided navigation, where external cues reduce reliance on internal maps. Importantly, eye movement metrics were matched across task conditions, suggesting that theta differences reflect cognitive demands rather than low-level visual or motor differences. Thus, MTL theta appears to integrate visual exploration with internal memory representations only when memory is required to guide behavior.

In summary, our findings show that eye movements modulate MTL theta oscillations during naturalistic navigation, particularly in memory-guided contexts. These results underscore the central role of vision in structuring memory processes in humans and reveal that saccades may provide temporal scaffolding for theta-mediated memory computations. Given that visual exploration and spatial memory are altered early in Alzheimer's disease and related disorders [50–55], saccade-linked theta may serve as a neural marker of memory function and a promising target for future interventions in both health and disease.

## Materials and methods

### Participants

Five individuals (ages 31–52; three male, two female) with pharmacoresistant focal epilepsy participated in this study. All participants had indwelling depth electrodes implanted in MTL regions for clinical purposes. Electrode placement was determined solely based on clinical criteria. All participants volunteered for the study and provided written informed consent in accordance with a protocol approved by the UCLA Medical Institutional Review Board (IRB #17-001452), and this study was conducted in accordance with the ethical principles for medical research involving human subjects as outlined in the Declaration of Helsinki. Data reported here were collected as part of a previously published study [16] and are reanalyzed to address novel hypotheses.

### Behavioral task

Participants performed a memory-guided spatial navigation task while freely moving within a rectangular room (approximately 5.9 × 5.2 m; ~19.4 × 17.1 ft). Twenty wall-mounted signs displaying color-number pairs served as navigational cues, and three predefined circular target locations (0.7 m diameter) were invisibly positioned within the space (Fig 1C). Trials alternated between two types: (1) visually-cued navigation, where participants walked to a visible wall sign (e.g., "yellow 4"); and (2) memory-cued navigation, where they searched for a previously learned but currently hidden target location (e.g., "T"). Hidden targets were never visually presented, either prior to or during the experiment. Instead, during initial trials, participants were instructed to explore until they reached the correct location, at which point an audio tone signaled that they were within the target's 0.7 m radius. This auditory feedback was repeated each time the target was reached, strengthening memory across trials. Each participant completed multiple blocks of alternating trial types (2–4 blocks, ~15 min each). The same hidden targets were repeated within a block (four for Participant 1; three for Participants 2–5), and the order of targets varied across blocks. During memory-cued trials, participants could either search freely or walk directly to the hidden target if recalled.

Performance on memory-cued trials was quantified using detour error, defined as the percent increase in walking distance relative to the optimal straight-line path. Trials were divided into low-, medium-, and high-performance trials by splitting the data into equal-sized thirds based on detour error scores. Participant 1 completed a slightly modified version of the task that included four hidden target locations. Additional observation-only task data from the original study were excluded from the present analysis. Full task details are described in Stangl and colleagues [16].

### Experimental control and data synchronization

Real-time experimental control was implemented using a custom Unity application (version 2018.1.9f2). Participant position and heading were continuously tracked via a motion capture system, and behavioral events (e.g., arrival at a target) triggered auditory instructions and task progression. The application automatically detected key events, such as reaching a wall sign or hidden target, and issued the next instruction accordingly. Synchronization signals were sent to the implanted iEEG acquisition system at the start and end of each 3.5–4 min recording segment to enable precise alignment with eye tracking and motion capture data. Technical details of the synchronization framework are provided in Topalovic and colleagues [9].

## Eye tracking and saccade detection

Mobile eye-tracking data were recorded using the Pupil Core headset (Pupil Labs GmbH) at ~200 Hz, with a spatial accuracy of 0.6°. Gaze position was calculated within a normalized 192 × 192 pixel reference frame. Data acquisition was performed using the Pupil Capture (v1.11) and processed with Pupil Player (v1.11). Eye movements were classified as saccades or fixations using the Cluster Fix toolbox for MATLAB [56], which applies k-means clustering to velocity, acceleration, and displacement metrics. Clusters with the lowest combined velocity and acceleration were labeled as fixations, along with any within three standard deviations of that cluster's mean. All remaining clusters were classified as saccades. Each time point was labeled accordingly and used throughout all analyses to distinguish neural activity during saccades versus fixations (S1A Fig). The distribution of saccade durations is shown in S1B Fig.

## Motion tracking

Participant location was continuously tracked throughout the experiment using the OptiTrack system (Natural Point, Corvalis, OR) and the MOTIVE software (version 2.2.0). Twenty-two wall-mounted infrared cameras captured the position of a rigid body marker, composed of multiple reflective markers affixed to the participant's head, at a sampling rate of 120 Hz with sub-millimeter precision. An additional camera from the system recorded a wide-angle video of the room for contextual reference during the experiment.

## iEEG data acquisition

iEEG was recorded using the US Food and Drug Administration (FDA)-approved RNS System (Neuropace, Mountain View, CA), which is typically used for seizure detection and responsive stimulation. To eliminate stimulation artifacts during the experiment, therapeutic stimulation was temporarily disabled with the participant's informed consent, and the system was used solely for recording. Each participant had two implanted depth leads, each with four electrode contacts spaced either 3.5 mm or 10 mm apart. This configuration enabled continuous bipolar iEEG recording from up to four channels per participant, sampling at 250 Hz. iEEG data were transmitted in real time via near-field telemetry using a Wand positioned over the implanted neurostimulator (Fig 1A). The Wand was mounted on a custom backpack worn by the participant, allowing free movement during recording (see Topalovic and colleagues [9] for technical details). Data were stored automatically in ~3.5–4 min segments, with brief (~10 s) breaks between recordings. During these breaks, participants were instructed via computerized audio prompts to remain still.

## Electrode localization

High-resolution post-operative head computed tomography (CT) scans was acquired for all participants. Electrode contact locations were determined by co-registering each participants CT to their preoperative high-resolution structural magnetic resonance image (T1- and/or T2-weighted), as shown in Fig 1B. Recording contacts were localized to MTL regions, including the hippocampus, subiculum, entorhinal cortex, parahippocampal cortex and perirhinal cortex (S2 Table). No contacts were located in the amygdala. Channels outside the MTL were excluded from further analyses.

## Detection of epileptic events

Epileptic events, including interictal epileptiform discharges (IEDs), were detected separately for each recording channel using a previously established method [57], which we have applied in prior RNS System studies [10,16,58]. The algorithm employs a double-thresholding approach based on two criteria (1) the envelope of the unfiltered signal exceeding six standard deviations above baseline, or (2) the envelope of the rectified signal filtered between 15 and 80 Hz exceeding the same threshold. To capture residual epileptic activity surrounding these events, the binary IED vector (1 for IED, 0 otherwise) was smoothed using a Gaussian filter with a 0.05 s window. All samples with smoothed values above 0.01 were

also marked as IEDs. This step effectively excluded a ~250 ms window around each detected IED. Using this method, an average of 1–2% of data per participant was identified as IED-related (range: 1.1% for participant 2% to 2.0% for participant 1), consistent with previous findings [10].

Participants were intentionally selected for relatively low IED activity. Using the RNS System's event log, we estimated each participant's average daily event count over the preceding three months and recruited those with fewer events (approximately 500–800 per day). All participants gave informed consent to disable stimulation during the experiment to eliminate stimulation-related artifacts in the iEEG recordings.

## iEEG data analysis

Time–frequency analyses of iEEG data were conducted using the BOSC toolbox [26,59], which includes both continuous wavelet-based power estimation and detection of oscillatory bouts based on duration and power criteria. For all main analyses, oscillatory power was computed using a continuous Morlet wavelet transform (implemented within the BOSC framework), which estimates time-resolved power at each frequency. Morlet wavelets of order six were applied across frequencies from 2 to 30 Hz in 0.25 Hz steps, and 31–80 Hz in 1 Hz steps. To quantify the prevalence of theta bouts, we additionally used the BOSC detection algorithm, which identifies time points where power exceeds the 95th percentile of a fitted background (1/$f$) distribution for at least two cycles; these results are reported in S1C Fig.

BOSC and wavelet analyses were applied to each 3.5–4 min recording interval on a sample-by-sample basis, providing high temporal resolution for linking transient neural dynamics to momentary behaviors such as saccadic eye movements. To ensure robustness, analyses were repeated with a lower cycle (3) wavelet approach, reducing frequency resolution while improving temporal precision. Results remained consistent, indicating that our findings were not dependent on this trade-off (S6 Fig).

Following wavelet decomposition, band power was calculated by averaging across defined frequency bands (e.g., 5–8 Hz for theta). Power time series were normalized by z-scoring each frequency band within each recording interval and channel. These normalized time series were aligned with eye-tracking labels (saccade or fixation), and average power was computed for samples of interest (e.g., all saccade-labeled samples).

To examine relationships between behavior and theta dynamics, we used linear mixed-effects models with normalized theta power as the response variable. Eye movement speed and body movement speed served as fixed-effect predictors, while recording channels were included as random effects to account for variability across channels and participants. Importantly, eye movements were modeled as a continuous variable (magnitude per time point), rather than as categorical saccade/fixation labels. All predictor variables were standardized prior to model fitting, allowing direct comparison of effect sizes. Standardized beta weights indicate the relative contribution of each predictor. Models were fitted separately for memory-cued and visually-cued navigation conditions (S4C Fig).

To further assess the relationship between MTL theta activity and saccade frequency or entropy, we segmented the continuous iEEG data into overlapping one-second windows with a 4 ms step size. For each window, we calculated the average theta power and the total number of saccades. Windows were then categorized into high- and low-saccade-frequency groups (or high- and low-saccade-entropy groups) based on a median split of the saccade count (or entropy) distribution within each participant and condition. Average theta power was computed separately for high- and low-frequency saccade windows within the memory-cued and visually-cued navigation conditions. This windowed analysis allowed us to examine how ongoing fluctuations in saccade frequency related to theta dynamics during different task conditions.

## Data subsampling for statistical analysis

Because participants freely walked around the environment during the experimental task, it is possible that they spent more time in one task condition (e.g., memory-cued versus visually-cued navigation, or vice versa) which would affect statistical comparisons (for example, comparisons of band power) between these conditions. Similarly, statistical

comparisons between saccades and fixation periods could be biased by differences in their prevalence or duration throughout the task. To ensure balanced comparisons, we performed all analyses on 500 iteratively generated, equally sized subsets of data. For each participant, we first determined the number of samples in each condition, then subsampled from the condition with more data to match the smaller condition. Parameters of interest (e.g., wavelet-based power estimates) were computed on each matched subset. This procedure was repeated 500 times, and results were averaged across iterations. This approach was applied consistently across both eye movement and navigation condition comparisons. All reported statistics and plots are based on these equal-sized, averaged datasets.

### Statistical testing

For participant-level analyses (e.g., S1D Fig), statistical comparisons were performed using one-sided permutation tests with 10,000 permutations, as all hypotheses were directional (e.g., expecting higher theta power during saccades versus fixations, or during memory-cued versus visually-cued navigation). For paired data, condition labels were shuffled, and the mean difference was recalculated repeatedly to build a null distribution. The observed mean difference was then tested against this distribution. To assess whether a single set of values (e.g., mean power) differed from zero, we randomly flipped the sign of each value, recalculated the mean, and repeated this 10,000 times to create a null distribution. The true mean was then compared to this distribution. In all permutation-based tests, p-values were computed by counting permutations with values greater than or equal to the observed statistic (i.e., including ties). This approach provides an accurate estimate of the probability of observing the test statistic under the null hypothesis and prevents underestimation of p-values. All channel-level analyses were implemented using the PALM framework [17], which constrains permutation exchangeability within each participant, allowing valid inference without assuming independence across electrodes. Multiple comparisons were corrected using the FDR method where applicable (e.g., Figs 3H and 4D) [60].

### Temporal analysis

To examine the timing of neural activity around saccade onset, we extracted oscillatory power from 100 ms before to 200 ms after each saccade. For normalization, baseline power was defined as the mean theta power across all fixation and saccade periods pooled across both task conditions. To isolate activity specific to the transition from fixation to saccade, we excluded pre-saccade data points (−100 to 0 ms) that overlapped with prior saccades, and post-saccade points (0 to +200 ms) that overlapped with subsequent fixations. As a result, the temporal plots (line graphs, heat maps) reflect the transition from fixation (−100 to 0 ms) to saccade onset and progression (0 to +200 ms). Because saccade durations varied across events (S1B Fig), timepoints further from saccade onset are more likely to overlap with fixations and were excluded, reducing the number of saccades contributing data at later timepoints. At each time point, we averaged oscillatory power across all qualifying saccades, separately for each recording channel. For heatmaps (e.g., Figs 2D, 3C, and 3F), we visualized oscillatory power across time (4 ms steps) and frequency (2–30 Hz in 0.25 Hz steps, 31–80 Hz in 1 Hz steps).

### Supporting information

**S1 Table. Number of completed trials and saccade and fixation counts per participant and condition.** Values indicate the total number of trials, saccades, and fixation periods included in the memory-cued (MC) and visually-cued (VC) conditions for each participant.
(S1_Table.XLSX)

**S2 Table. Channel localization across participants.** Localization of two electrode contacts (+/−) per bipolar recording channel. HPP, hippocampus (CA1/CA2/CA3/DG); PRC, perirhinal cortex; PWM, parahippocampal white matter; Sub, subiculum; ERC, entorhinal cortex.
(S2_Table.XLSX)

**S1 Fig. Saccade characteristics and oscillatory prevalence. (A)** Example two-second segment showing two-dimensional eye velocity (red line) and the corresponding saccade index (black line). **(B)** Distribution of saccade durations (ms) and fixation durations (ms) across all participants ($n_{participants}=5$). **(C)** Prevalence of low-frequency oscillations shown as a percentage (%) of total samples. Oscillatory bouts were detected for individual frequency steps between 4 and 25 Hz during memory-cued and visually-cued navigation. Shaded gray area represents SEM across channels ($n_{channels}=16$). **(D)** No significant differences in saccade metrics between memory-cued (MC) and visually-cued (VC) conditions: mean duration ($p=0.47$), frequency ($p=0.15$), displacement ($p=0.31$), or fixation duration ($p=0.18$). **(E)** Similarly, saccade met-rics did not differ across performance levels. Mean saccade duration, frequency, displacement, and fixation duration did not differ significantly between HP, MP, and LP trials (all $p>0.13$). n.s. = not significant. The data underlying this Figure are available here: https://doi.org/10.5281/zenodo.18487389.
(S1_Fig.TIF)

**S2 Fig. Saccade-related theta modulation differs between planning and execution phases of memory-cued trials. (A–C)** For memory-cued (MC) navigation, each trial was divided into equal first-half (planning) and second-half (execu-tion) epochs. Normalized (z-scored) theta (5–8 Hz) activity was significantly higher during saccades compared to fixations in the first half (* $p=0.013$; **A**), but not in the second half (n.s., $p=0.254$; **B**). The difference in saccade-fixation theta mod-ulation between halves was significant (* $p=0.041$; **C**). **(D–F)** In the visually-cued (VC) navigation condition, no saccade–fixation differences were observed in the first half (n.s., $p=0.546$; **D**), second half (n.s., $p=0.327$; **E**) or their difference (n.s., $p=0.772$; **F**). **(G–H)** Movement speed increased from planning to execution in both MC navigation (* $p=0.030$; **G**) and visually-cued navigation (* $p=0.031$; **H**), confirming greater locomotor engagement in the latter half of each trial. n.s. = not significant. The data underlying this Figure are available here: https://doi.org/10.5281/zenodo.18487389.
(S2_Fig.TIF)

**S3 Fig. Theta phase resetting to saccade onset. (A–B)** Theta phase resetting relative to saccade onset, quantified using pairwise phase consistency (PPC). PPC was significantly elevated from −60 to +196 ms around saccade onset (black bar = $p<0.05$, shaded areas represent SEM across participants; **A**), and PPC during the saccade-onset window (−50 to +200 ms) was significantly higher than an equivalently sized baseline period (−800 to −550 ms, ** $p<0.01$; **B**). **(C–F)** Time-resolved PPC traces (mean ± SEM across participants, shaded gray) show significant theta phase alignment surrounding saccade onset in both memory-cued (MC) and visually-cued (VC) navigation. **(G–J)** When trials were divided into early and late halves, significant PPC increases were observed around saccade onset in both periods, indicating con-sistent phase resetting throughout the trial. Bar plots adjacent to each panel display mean PPC during the saccade-onset window (−50 to +200 ms) and a baseline window (−800 to −550 ms). PPC was significantly higher during the saccade period for all comparisons (** $p<0.01$). These results demonstrate that theta phase alignment accompanies saccadic events across both task conditions and throughout the course of navigation. The data underlying this Figure are available here: https://doi.org/10.5281/zenodo.18487389.
(S3_Fig.TIF)

**S4 Fig. Saccades associated with memory-related theta oscillations. (A)** MTL low-frequency power was elevated during memory-cued navigation compared to visually-cued navigation within the 2.0–3.75 Hz and 6.75–17.5 Hz ranges ($p<0.05$). **(B)** The saccade-related oscillatory effect, characterized by an increase in band power during saccades, was more pronounced during memory-cued than during visually-cued navigation, particularly in the theta frequency range (5.75–7.5 Hz). **(C)** A linear mixed-effects model was employed to assess the concurrent impact of eye and body move-ment on theta band power, separately for memory- and visually-cued navigation. In memory-cued navigation, eye move-ment speed significantly contributed to theta power ($p=0.032$), whereas body movement speed did not ($p=0.818$). In visually-cued navigation, neither eye movement speed ($p=0.436$) nor body movement speed ($p=0.656$) appeared to modulate changes in theta power. **(D)** During memory-cued navigation, theta (5–8 Hz) power in the MTL trended higher

during high- versus low-saccade frequency periods ($p = 0.096$). During visually-cued navigation, theta power in the MTL did not significantly differ between high- and low-saccade frequency periods ($p = 0.630$). **(E)** Comparison of the high–low entropy theta difference between memory-cued and visually-cued conditions revealed a significant interaction ($p = 0.049$). For **A,B** shaded gray = standard error of the mean (SEM) across channels ($n_{channels} = 16$). Black horizontal bars = $p < 0.05$. **C** shows mean (± SEM) across 5 participants (circles). **D,E** shows mean (± SEM) across 16 channels (circles). ♦ = p < 0.1, * = $p < 0.05$, n.s. = not significant. The data underlying this Figure are available here: https://doi.org/10.5281/zenodo.18487389.
(S4_Fig.TIF)

**S5 Fig. Control analyses for saccade-related modulation of theta band power.** Normalized (z-scored) MTL theta power during saccades is shown separately for memory-cued and visually-cued conditions, in addition to direct comparisons between memory-cued and visually-cued conditions, under four behavioral contexts: **(A-B)** near boundaries (<1.2 m from the wall), **(C-D)** inner region (>1.2 m from the wall), **(E-F)** during movement, and **(G-H)** while stationary. Saccade-related theta increases were observed primarily during memory-cued navigation, with a significant effect near boundaries and trends in the other contexts. Each point reflects a single MTL channel (N = 16); bars represent mean ± SEM. ♦ = p < 0.1, ** = p < 0.01, n.s. = not significant. The data underlying this Figure are available here: https://doi.org/10.5281/zenodo.18487389.
(S5_Fig.TIF)

**S6 Fig. Analyses with increased temporal resolution.** All primary analyses were repeated using BOSC parameters optimized for higher temporal resolution at the expense of frequency resolution (see Materials and Methods). The results remained consistent with the main findings. **(A)** MTL low-frequency (3–15 Hz) power was significantly higher during saccades than fixations ($p = 0.001$), replicating Fig 2B. **(B)** During memory-cued (MC) navigation, MTL theta (5–8 Hz) power increased during saccades relative to fixations ($p = 0.013$), replicating Fig 3A; no significant theta increase was observed during visually-cued (VC) navigation ($p = 0.258$), consistent with Fig 3D. **(C)** Within MC trials, theta power increases were stronger during high-performance (HP; $p = 0.012$) and medium-performance (MP; $p = 0.035$) trials compared to low-performance (LP) trials, consistent with Fig 3H. **(D)** Theta power was higher during high- versus low-entropy periods in MC navigation ($p = 0.023$) but not during VC navigation ($p = 0.158$), replicating Fig 4E and 4F. **(E)** Theta power was higher during long versus short saccades in MC navigation ($p = 0.004$) but not during VC navigation ($p = 0.205$), replicating Fig 4B. **(F)** Theta power showed a trend toward being higher during high- versus low-frequency saccades in MC navigation ($p = 0.096$) but not during VC navigation ($p = 0.630$), replicating S4D Fig. **(G)** Theta power was higher during saccades compared to fixations in boundary regions during MC navigation ($p = 0.007$) but not during VC navigation ($p = 0.506$), replicating S5A Fig. **(H)** Theta power did not differ between saccades and fixations in inner-room regions for either MC or VC navigation (MC, $p = 0.130$; VC, $p = 0.164$), replicating S5C Fig. **(I)** Theta power was higher during saccades compared to fixations during the first half of MC trials ($p = 0.013$) but not during the second half ($p = 0.251$), replicating S2A and S2B Fig. All panels show mean ± SEM across 16 MTL channels (circles). * $p < 0.05$, ** $p < 0.01$, ♦ $p < 0.1$; n.s. = not significant. The data underlying this Figure are available here: https://doi.org/10.5281/zenodo.18487389.
(S6_Fig.TIF)

**S1 Video. Representative trials showing simultaneous iEEG and eye tracking during navigation.** Example trials from memory-cued (left) and visually-cued (right) navigation are shown. Video displays include: (Top left) eye-facing head-mounted cameras capturing both eyes; (Top) world-facing head-mounted camera showing the participant's first-person view with real-time gaze position (yellow circle). (Bottom) plots show (top trace) normalized theta band power from a representative MTL channel and (bottom trace) raw iEEG signal, time-aligned to the video. Right panel shows eye speed computed using a 2 s moving average window with one-sample increments.
(S1_Video.MP4)

## Acknowledgments

The authors thank all members of the Suthana Lab for helpful discussions and the participants for taking part in the study.

## Author contributions

**Conceptualization:** Humza N. Zubair, Matthias Stangl, Nanthia Suthana.

**Data curation:** Humza N. Zubair, Matthias Stangl.

**Formal analysis:** Humza N. Zubair, Matthias Stangl.

**Funding acquisition:** Matthias Stangl, Nanthia Suthana.

**Investigation:** Matthias Stangl, Sonja Hiller.

**Methodology:** Matthias Stangl, Uros Topalovic.

**Project administration:** Matthias Stangl, Sonja Hiller, Dawn Eliashiv, Nanthia Suthana.

**Resources:** Uros Topalovic, Vikram R. Rao, Casey H. Halpern, Dawn Eliashiv, Itzhak Fried, Nanthia Suthana.

**Software:** Humza N. Zubair, Matthias Stangl, Uros Topalovic, Martin Seeber.

**Supervision:** Matthias Stangl, Uros Topalovic, Vikram R. Rao, Nanthia Suthana.

**Validation:** Matthias Stangl.

**Visualization:** Humza N. Zubair, Uros Topalovic.

**Writing – original draft:** Humza N. Zubair, Matthias Stangl, Nanthia Suthana.

**Writing – review & editing:** Humza N. Zubair, Matthias Stangl, Uros Topalovic, Cory Inman, Martin Seeber, Sonja Hiller, Vikram R. Rao, Casey H. Halpern, Dawn Eliashiv, Itzhak Fried, Nanthia Suthana.

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
