## [Editor Report · Decision Letter 0]

9 Jul 2025

Dear Dr Suthana,

Thank you for submitting your manuscript entitled "Eye Movements Reveal Memory-Related Theta Activity in the Human Brain" for consideration as a Short Reports by PLOS Biology.

Your manuscript has now been evaluated by the PLOS Biology editorial staff and I am writing to let you know that we would like to send your submission out for external peer review.

Once your full submission is complete, your paper will undergo a series of checks in preparation for peer review. After your manuscript has passed the checks it will be sent out for review. To provide the metadata for your submission, please Login to Editorial Manager (https://www.editorialmanager.com/pbiology) within two working days, i.e. by Jul 11 2025 11:59PM.

Kind regards,

Christian

Christian Schnell, PhD,

Senior Editor

PLOS Biology

cschnell@plos.org

---

## [Decision Letter · Decision Letter 1]

3 Sep 2025

Dear Dr Suthana,

Thank you for your patience while your manuscript "Eye Movements Reveal Memory-Related Theta Activity in the Human Brain" was peer-reviewed at PLOS Biology. It has now been evaluated by the PLOS Biology editors, an Academic Editor with relevant expertise, and by several independent reviewers.

In light of the reviews, which you will find at the end of this email, we would like to invite you to revise the work to thoroughly address the reviewers' reports.

As you will see below, the reviewers agree that your study is very interesting, but raise some concerns about the statistical analyses, missing analyses to provide stronger support for the claims and the need for better contextualization with the existing literature.

Given the extent of revision needed, we cannot make a decision about publication until we have seen the revised manuscript and your response to the reviewers' comments. Your revised manuscript is likely to be sent for further evaluation by all or a subset of the reviewers.

**IMPORTANT - SUBMITTING YOUR REVISION**

*Re-submission Checklist*

*Published Peer Review*

*PLOS Data Policy*

Sincerely,

Christian

Christian Schnell, PhD

Senior Editor

PLOS Biology

cschnell@plos.org

REVIEWS:

Reviewer #1 (Jack J. Lin, signed his report): Summary and Overall Assessment

This manuscript presents an elegant and technically sophisticated investigation into the relationship between saccadic eye movements and medial temporal lobe (MTL) theta activity during ambulatory spatial navigation in humans. Using mobile intracranial EEG recordings from five patients with implanted RNS devices, combined with high-resolution motion tracking and eye-tracking, the authors demonstrate that theta power increases during saccades—but selectively so under memory-guided conditions. Moreover, these saccade-related theta increases are most pronounced during trials with better memory performance, suggesting a functional link between visual sampling and memory success.

This study offers a compelling framework to reconcile interspecies differences in theta dynamics by highlighting how humans use vision-based active sensing (saccades) in ways that may parallel whisking in rodents. The authors' interpretations are well-argued and thoughtfully positioned within the existing literature. The analytical rigor, creative use of mobile iEEG technology, and translational relevance of the findings make this a valuable contribution.

Below, I outline the major strengths, followed by two substantive suggestions for improvement and several minor comments.

Major Strengths

1. Innovative Methodology:

The authors leverage state-of-the-art mobile iEEG to simultaneously record hippocampal activity, body movement, and eye tracking in freely moving humans—an impressive technical achievement that allows for ecologically valid cognitive neuroscience.

2. Cognitive Specificity:

By contrasting memory-cued with visually guided navigation and matching oculomotor metrics across these conditions, the study isolates the cognitive context (memory demand) as the key modulator of saccade-related theta increases.

3. Behavioral Relevance:

The link between theta power and successful memory trials (based on detour distance) provides meaningful evidence that theta is not merely movement- or attention-related but behaviorally significant.

4. Cross-Species Integration:

The discussion thoughtfully relates the findings to species differences in active sensing, offering a unifying framework for understanding theta across rodents, non-human primates, and humans.

5. Analytical Rigor:

The authors apply robust and transparent statistical methods—including leave-one-subject-out validation, permutation testing, and sensitivity checks across time-frequency resolutions—demonstrating excellent methodological care.

Major Suggestions for Improvement

1. Disentangling Planning from Memory Retrieval

While the authors interpret saccade-theta coupling as a signature of memory engagement, it is equally plausible that these effects reflect visuospatial planning demands, which are inherent in the memory-cued condition. Participants must retrieve the target location but also plan a route to an unseen goal, engaging additional cognitive operations distinct from retrieval per se.

To strengthen this interpretation, the authors could consider:

* Segmenting trials into pre-movement (planning) and in-motion (execution) epochs to compare theta modulation across these phases.

* Categorizing saccades as goal-congruent or non-congruent, and testing whether theta is more strongly modulated by those aligned with future path trajectories.

* Computing scanpath entropy or dispersion metrics to quantify planning load, and examining how they relate to theta power.

If theta increases are stronger during structured, anticipatory eye movement patterns or early trial epochs, this would support a broader role for theta in prospective visual planning, beyond memory recall alone.

2. Theta Phase Resetting and Temporal Alignment

The manuscript focuses on theta amplitude changes, but prior studies in both humans and non-human primates suggest that saccades can induce phase resetting in MTL theta. Such phase alignment may temporally organize neural ensembles for memory encoding and retrieval, and could be present even in the absence of large amplitude changes.

I encourage the authors to explore:

* Whether theta phase aligns to saccade onset, using measures like inter-trial phase coherence or pairwise phase consistency.

* Whether phase resetting varies by task condition or navigation phase (e.g., stronger near goal locations or early in trials).

* Whether trial-by-trial phase consistency predicts behavioral performance (e.g., detour error), thereby complementing the amplitude-based findings.

Such analyses could provide a more mechanistic account of how saccades organize memory-relevant computations in the MTL.

Reviewer #2: The authors examine the relationship between eye movements, ambulatory motion, and MTL theta power in intracranial EEG recordings from five freely moving patients performing a spatial navigation task. They find that MTL theta power is increased during saccades, specifically during memory guided navigation, that this correlates with navigational accuracy, and is greater for longer saccade distances. These are rare data, the results are novel and should be of interest to the wider neuroscience community. I just have a few concerns about the statistical analyses presented here, and require clarification on a few aspects of the empirical methods.

Major Comments

It is not clear to me - from either the main text, Figure 1 caption, or Methods - how participants learned the invisible target locations?

At various junctures, the authors present statistical results across channels alongside results across patients. This implicitly assumes that data from each iEEG electrode contact can be treated as an independent observation, which is clearly not the case. As such, all statistical analyses across channels should be removed from the manuscript. As far as I can tell, this will not affect the overall pattern results at all, as the across channel / across patient statistical results are always consistent

On a related note, please report the values of test statistics and degrees of freedom (or simply, n=5) for all statistical tests throughout the manuscript. In places (e.g. when examining whether 'differences in saccadic modulation of theta power could be explained by variations in eye movement behavior across conditions') this information is not given, and so it is not possible to ascertain whether these analyses are across channels or across participants. It should also be made clear in the main text whether test results are from one- or two- sided tests

In Figures 2C and D, theta power begins to increase shortly (~50ms) after, and continues to increase up to 200ms after, saccade onset. I appreciate that "saccade durations varied across events" (with a median value of ~90ms), and that later time points are more likely to overlap with fixations, but it would still be interesting to know how long this theta power increase persisted for (particularly for the memory-cued condition)

At the top of page 8, the authors ask whether saccade related increases in theta power can be explained by other behavioural variables - such as distance from boundaries, body movement etc. In each case, they compare p-values for tests of theta power in each condition. Instead (or in addition to this), they should directly compare saccade related theta power in each condition using a paired t-test (or equivalent) and report those results. In addition, it is not clear to me why separate linear models were fit for each condition - why not have a single mixed linear model and look for any significant interaction?

Finally, I wonder if the authors wish to speculate on the implication of these results for previous reports of movement related MTL theta power. Is it possible that participants make more saccades during movement (when the visual scene changes) than during stationary periods (perhaps the authors could test that in their data?), and that previous reports of more prevalent theta oscillations during movement could therefore be explained by increased saccade frequency, rather than translational movement per se?

Minor Comments

Introduction: "This contrasts with the approach of directly recording brain activity during freely-moving ambulatory behavior in rodents" - this seems like a repeat of the information in the previous sentence, the authors might consider cutting?

Results: "We analyzed mobile iEEG recordings in five individuals who were surgically implanted with the RNS System … In addition, we examined eye and body movements captured while participants were engaged in a spatial navigation task within a rectangular room" - this makes it sounds as if these were two separate cohorts or analyses - perhaps it would be better to say "Alongside the EEG signals, we examined eye and body…" ?

Methods: "via computerized to remain still" - possible typo?

Methods: The iEEG data analysis section is confusing, because it briefly describes the BOSC toolbox and method, but as far as I can tell, all results presented in the manuscript use oscillatory power as estimated by a wavelet transform, rather than the prevalence of theta bouts as estimated by BOSC. Can the authors please clarify?

Figure 4: Do these figures all have y-axis units of Z - i.e. report average Z-scored data? If so, please add the units accordingly (this is more specific than 'normalised' power, which is fairly ambiguous)

Methods: It would be useful if the authors could report how many trial numbers in each condition were completed by each participant, how many fixation periods were included, and show the distribution of fixation durations (i.e. to match the data for saccades presented throughout the manuscript)

Reviewer #3: Zubair et al. report the result of the analysis of simultaneously recorded hippocampal LFP and eye movements in freely-moving human participants, during the performance on memory-guided and visually-guided navigation. They report higher theta power (5-8 Hz) during saccadic eye movements, relative to fixation periods, selectively during the trials with memory demands. In addition, hippocampal theta power was associated with spatial memory performance and higher amplitude saccades. The results are interesting and contribute to understanding the primate-specific relation between the visual sampling, memory and navigation. However, some of the aspects of analysis and interpretation deserve scrutiny and thorough clarification, before the paper could be deemed acceptable for publication.

Major points:

The fixation baseline used for contrasting the saccadic vs. fixation theta is -100 - 0 ms, relative to saccadic onset. However, as evident from Fig. 2B, theta power increases over the baseline around 50ms post-saccadic onset and keeps ramping up until 250ms. The crucial aspect missing from this analysis is the time course of theta power after the saccade offset? Since the fixations are typically quite longer than the last 100ms used for this comparison, the baseline used in the analysis might result in generalizing the theta power just prior to saccade onset as characteristic for the entire fixation period.

If the theta power remains elevated during the early fixation period, it's hard to argue that the theta is increased during saccades, relative to fixation periods. This question should be clarified by showing the analysis relative to saccade offset, as well. The statement of higher theta during saccade, relative to fixation, should be re-examined in the context of this analysis.

As shown in multiple previous publications (e.g. Katz et al., 2022, Fig. 4A), peri-saccadic hippocampal LFP is dominated by the ERP with early peak starting around the same time as the theta increase during saccades in the current study (50 ms post saccadic onset; Fig. 2C in Katz et al., 2020). The authors should acknowledge that the appearance of theta (or broader low-frequency power) in peri-saccadic spectrograms could also originate from this activity. In addition, these previous studies in humans (Katz et al., 2020, 2022) and non-human primates (Hoffman et al., 2013) showed that this peri-saccadic ERP / transient theta is present during visual tasks that do not put demands on memory as in the current analyses. Other work links saccade-locked theta during encoding to subsequent memory (Jutras, Fries & Buffalo, 2013), while the current reported results seem to suggest these signals are present for memory retrieval. Thus, a more in depth treatment of these discrepancies and why the current results deviate from prior literature is warranted. Furthermore, some of these papers emphasized theta phase resetting more than power changes per say. Thus, it is possible that theta phase was reset during non-memory related tasks, even if there was no commensurate change in power. Thus, follow up phase consistency analyses should be performed to rule out this possibility, to better fit with the extant literature. This is especially important considering the theoretical importance of theta phase for memory-perception interactions (Hasselmo, Bodelón & Wyble, 2002; Lisman & Buzsáki, 2008).

Given the median saccade length of 62 ms, the theta doesn't seem to show any increase from baseline for a large fraction of saccades. The results seem to be driven mostly by a long tail of saccades longer than 100 ms. This observation should receive more in depth treatment, when contextualizing the interpretation of the current findings.

It's difficult to conceptualize that the theta power increases for a relatively small fraction of an oscillatory cycle, which would result either in a highly asymmetric oscillatory waveform, which could again be interpreted as a nested higher frequency oscillations showing the phase-amplitude coupling with underlying theta. The authors do specify that the BOSC toolbox detects oscillations based on duration (greater than or equal to two cycles), and their analyses were also robust even after using wavelets with 3 cycles instead of 6. However, one limitation of wavelet-based signal processing approaches is that, even with a low number of cycles that maximize temporal resolution, they induce temporal smoothing in the signal over time, which can inflate oscillation duration estimates present in the time domain. Thus, it would be informative to see if multicycle rhythms are observed when evaluating rhythms using non-spectral oscillation analysis methods performed in the time-domain, like cycle-by-cycle analysis for example (Cole & Voytek, 2019).

Finally, as acknowledged by the authors (Lines 281-286), previously published results from the same group show the correlation between theta power and body movement speed, which is missing in hereby presented results. This should be clarified.

Minor points:

It's unclear if the trial type (memory vs. visually-guided) was alternated following each trial or if there were longer blocks of the same trial type.

Were the subjects given any feedback about how close their final position was to the invisible target?

Were the different targets randomly selected in consecutive memory trials or there were blocks when the participants searched for the same targets?

It is unclear how the trials were divided into low-, medium- and high-performance. Was it based on clustering or data points split into equal-sized thirds or some other objective method?

It would be interesting to know if the theta during saccades increased specifically during saccades that landed on the target location.

References

Hasselmo, M. E., Bodelón, C., & Wyble, B. P. (2002). A proposed function for hippocampal theta rhythm: separate phases of encoding and retrieval enhance reversal of prior learning. Neural computation, 14(4), 793-817.

Lisman, J., & Buzsáki, G. (2008). A neural coding scheme formed by the combined function of gamma and theta oscillations. Schizophrenia bulletin, 34(5), 974-980.

Hoffman, K. L., Dragan, M. C., Leonard, T. K., Micheli, C., Montefusco-Siegmund, R., & Valiante, T. A. (2013). Saccades during visual exploration align hippocampal 3-8 Hz rhythms in human and non-human primates. Frontiers in systems neuroscience, 7, 43.

Katz, C. N., Patel, K., Talakoub, O., Groppe, D., Hoffman, K., & Valiante, T. A. (2020). Differential generation of saccade, fixation, and image-onset event-related potentials in the human mesial temporal lobe. Cerebral Cortex, 30(10), 5502-5516.

Katz, C. N., Schjetnan, A. G., Patel, K., Barkley, V., Hoffman, K. L., Kalia, S. K., ... & Valiante, T. A. (2022). A corollary discharge mediates saccade-related inhibition of single units in mnemonic structures of the human brain. Current Biology, 32(14), 3082-3094.

Jutras, M. J., Fries, P., & Buffalo, E. A. (2013). Oscillatory activity in the monkey hippocampus during visual exploration and memory formation. Proceedings of the National Academy of Sciences, 110(32), 13144-13149.

Cole, S., & Voytek, B. (2019). Cycle-by-cycle analysis of neural oscillations. Journal of neurophysiology, 122(2), 849-861.

---

## [Decision Letter · Decision Letter 2]

15 Jan 2026

Dear Dr Suthana,

Thank you for your patience while we considered your revised manuscript "Eye Movements Reveal Memory-Related Theta Activity in the Human Brain" for consideration as a Short Reports at PLOS Biology. Your revised study has now been evaluated by the PLOS Biology editors, the Academic Editor and two of the original reviewers.

In light of the reviews, which you will find at the end of this email, we are pleased to offer you the opportunity to address the remaining points from Reviewer 3 in a revision that we anticipate should not take you very long. We will then assess your revised manuscript and your response to the reviewers' comments with our Academic Editor aiming to avoid further rounds of peer-review, although we might need to consult with the reviewers, depending on the nature of the revisions.

**IMPORTANT - SUBMITTING YOUR REVISION**

*Resubmission Checklist*

*Published Peer Review*

*PLOS Data Policy*

*Blot and Gel Data Policy*

Sincerely,

Christian

Christian Schnell, PhD

Senior Editor

PLOS Biology

cschnell@plos.org

REVIEWS:

Reviewer #1 (Jack J. Lin signed his report): I appreciate the authors' thoughtful, and technically rigorous responses to my comments. They have comprehensively addressed my concerns with substantial new analyses that significantly strengthen the manuscript and sharpen its conceptual contributions.

In particular, the new analyses dissociating planning and execution phases provide important clarification. Demonstrating that saccade-related theta modulation is strongest during the early, planning-dominant phase of memory-guided navigation extends the interpretation beyond memory retrieval alone. The addition of scan path entropy analyses further reinforces this conclusion by linking increased theta power to more exploratory and dispersed gaze patterns specifically during memory-guided navigation. Together, these findings position theta as a neural signature of internally driven planning during active sensing, broadening its functional role beyond memory per se.

Equally important, the demonstration of robust theta phase resetting aligned to saccade onset, present across both memory-guided and visually guided conditions and throughout trial epochs, supports the idea of saccades acting as temporal anchors for MTL network coordination. The fact that phase resetting occurs in both paradigms argues for a general organizing mechanism coupling visual sampling with hippocampal timing.

Overall, the authors have responded constructively and comprehensively, adding multiple converging analyses that elevate both the rigor and the interpretive clarity of the work. The revised manuscript represents an important advancement in our understanding of how eye movements, theta dynamics, and internal cognitive processes such as planning and memory interact during naturalistic human behavior.

Reviewer #2: Results (relating to my previous comment C3): It might be clearer if the description of the PALM framework is moved one sentence earlier in the Results, so that it appears just before the first statistical result that is generated using that framework, eg: "Channel-level statistics across the five participants were computed using the Permutation Analysis of Linear Models (PALM) framework [17], which constrains permutations within participants. This approach maintains appropriate statistical independence while preventing effects from being driven by any single individual. Because PALM operates at the level of individual channels while preserving participant structure, we report channel-level analyses throughout the manuscript. Using this approach, we observed a significant increase in MTL power within the low-frequency range (~3-15Hz) during saccades compared to fixation periods when aggregating across channels (Nchannels = 16; t = 3.71, p < 0.001; Figure 2A-B). Additionally, we found that low-frequency power rose gradually following saccade onset..."

Results (relating to my previous comment C5): The various findings relating saccades to theta power remain difficult to interpret - specifically, saccade related theta power during memory guided navigation trials is higher: (i) when participants are close to the boundaries; (ii) during high or medium performance trials; (iii) during the first half of each trial ('planning' period); and (iv) when saccades are faster, longer or more complex. As I described previously, the problem is that each of these contrasts have been performed separately, rather than within a single statistical framework, so it is not clear whether these predictors are correlated or confounded. For example, do memory guided trials tend to start near the boundaries (following navigation to a visual landmark on the wall), meaning that the early phases of navigation tend to take place near the boundaries, explaining (i) and (iii)? Do high and medium performance trials tend to involve faster or more complex saccade patterns, explaining (ii) and (iv)? I am aware that the authors compared saccade duration, frequency, and displacement between trials with different levels of performance - but not speed or complexity. In any case, I appreciate that a linear mixed-effects model may not be suitable in this case, due to the limited sample size, but perhaps the authors could directly examine the relationship between the predictors themselves instead? i.e. can they check whether more time is spent close to the boundaries, or whether saccades are faster, longer, or more complex during the first vs second half of trials, and so on? This might help to clarify the key behavioural parameters that dictate saccade related theta power increases

Also I believe the key for the significance values in the caption of Figure S5 is wrong: ** is not defined, and the diamond should be p<0.1 rather than p<0.01?

Results: "standard wavelet approach with a wavenumber of 6" - it might be clearer for a general audience if the authors refer to a "six cycle wavelet approach", as 'wavenumber' is simply a variable name in a specific implementation of the wavelet transform (ie the BOSC toolbox), and I'm not sure there's anything particular standard about choosing a six cycle wavelet, it just happens to be the default parameter in that toolbox. In addition, it would be better to replace 'wavenumber' with 'number of cycles' throughout the remainder of this section. Finally, for the sake of completness, the authors should also report the results of the other contrasts listed above (i, iii, iv) when a shorter wavelet is used, whether or not they are significant

Discussion: "Importantly, this saccade-theta relationship persisted regardless of the participant's position..." - is this statement true? In the results, the authors show that the saccade-theta relationship is not present in the inner regions of the environment. Please clarify

---

## [Editor Report · Decision Letter 3]

11 Feb 2026

Dear Nanthia,

Thank you for your patience while we considered your revised manuscript "Eye Movements Reveal Memory-Related Theta Activity in the Human Brain" for publication as a Short Reports at PLOS Biology. This revised version of your manuscript has been evaluated by the PLOS Biology editors and the Academic Editor.

Based on our Academic Editor's assessment of your revision, we are likely to accept this manuscript for publication, provided you satisfactorily address the following data and other policy-related requests:

* We would like to suggest a different title to improve its accessibility for our broad audience. Would either of this work for you?

Memory-related theta oscillation dynamics in the human brain are tightly coupled to eye movements during memory-guided navigation

OR

Eye movements reflect memory-related theta activity in the human brain

* Please include information in the Methods section whether the study has been conducted according to the principles expressed in the Declaration of Helsinki.

* DATA POLICY:

Regardless of the method selected, please ensure that you provide the individual numerical values that underlie the summary data displayed in the following figure panels as they are essential for readers to assess your analysis and to reproduce it: 2B, 3ADGH, 4 (all panels), S1DE, S2 (all panels), S3BDFHJ, S4CDE, S5 (all panels) and S6 (all panels).

* CODE POLICY

Per journal policy, if you have generated any custom code during the course of this investigation, please make it available without restrictions. Please ensure that the code is sufficiently well documented and reusable, and that your Data Statement in the Editorial Manager submission system accurately describes where your code can be found. More information on our Code Policy, what and how to share can be found here: https://journals.plos.org/plosbiology/s/code-availability

We expect to receive your revised manuscript within two weeks.

*Published Peer Review History*

*Press*

Sincerely,

Christian

Christian Schnell, PhD

Senior Editor

cschnell@plos.org

PLOS Biology

---

## [Editor Report · Decision Letter 4]

23 Feb 2026

Dear Nanthia,

Thank you for the submission of your revised Short Reports "Eye Movements Reflect Memory-Related Theta Activity in the Human Brain" for publication in PLOS Biology. On behalf of my colleagues and the Academic Editor, Raphael Kaplan, I am pleased to say that we can in principle accept your manuscript for publication, provided you address any remaining formatting and reporting issues. These will be detailed in an email you should receive within 2-3 business days from our colleagues in the journal operations team; no action is required from you until then. Please note that we will not be able to formally accept your manuscript and schedule it for publication until you have completed any requested changes.

While you attend to those requests, please also add the link to the source data in all corresponding figure legends in the main manuscript file and the supplementary information.

PRESS

Sincerely,

Christian

Christian Schnell, PhD

Senior Editor

PLOS Biology

cschnell@plos.org